# Ruptured Recurrent Interstitial Ectopic Pregnancy Successfully Managed by Laparoscopy

**DOI:** 10.3390/diagnostics14050506

**Published:** 2024-02-27

**Authors:** Claudiu Octavian Ungureanu, Floris Cristian Stanculea, Niculae Iordache, Teodor Florin Georgescu, Octav Ginghina, Raul Mihailov, Ileana Adela Vacaroiu, Dragos Eugen Georgescu

**Affiliations:** 1Department of Surgery, “Carol Davila” University of Medicine and Pharmacy, 37, Dionisie Lupu Street, 020021 Bucharest, Romania; claude.hack@gmail.com (C.O.U.); floris-cristian.stanculea@drd.umfcd.ro (F.C.S.); niordache@gmail.com (N.I.); octavginghina@gmail.com (O.G.); gfdragos@yahoo.com (D.E.G.); 2General Surgery Department, “Sf. Ioan” Clinical Emergency Hospital, 13 Vitan-Bârzeşti Road, 042122 Bucharest, Romania; 3General Surgery Department, Clinical Emergency Hospital Bucharest, SplaiulIndependentei nr 168, Sect 5, 050098 Bucharest, Romania; 4General Surgery Department, “Prof. Dr. Alexandru Trestioreanu” Oncological Institute, 022328 Bucharest, Romania; 5General Surgery Department, “Sf. Andrei” Clinical Emergency Hospital, 177 Braila Street, 800578 Galati, Romania; raul.mihailov@ugal.ro; 6Nephrology and Dialysis Clinic, “Sf. Ioan” Emergency Clinical Hospital, Sos. Vitan-Barzesti nr 12, 042122 Bucharest, Romania; 7General Surgery Department, “Dr. I. Cantacuzino” Clinical Hospital, 5-7 I. Movila Street, 022904 Bucharest, Romania

**Keywords:** ectopic pregnancy, hemorrhagic emergency, interstitial pregnancy, laparoscopic cornual resection

## Abstract

Ectopic pregnancies are a frequently encountered cause of first-trimester metrorrhagia. They occur when an embryo is implanted and grows outside the normal uterine space. Uncommonly, the embryo can be implanted in the intramural portion of the uterine tube, a condition referred to as interstitial localization. This specific type of ectopic pregnancy may have an unpredictable course, potentially leading to severe uterine rupture and catastrophic bleeding if not promptly diagnosed and managed. We present a rare case of a multiparous 36-year-old female patient who underwent pelvic ultrasonography in the emergency department for intense pelvic pain associated with hypotension and moderate anemia. A history of right salpingectomy for a ruptured tubal ectopic pregnancy 10 years previously was noted. High beta-HCG levels were also detected. A pelvic ultrasound allowed us to suspect a ruptured ectopic interstitial pregnancy at 8 weeks of amenorrhea. An association with hemoperitoneum was suspected, and an emergency laparoscopy was performed. The condition was confirmed intraoperatively, and the patient underwent a right corneal wedge resection with suture of the uterine myometrium. The postoperative course was uneventful, and the patient was discharged on the fourth day postoperatively. Interstitial ectopic pregnancy is a rare yet extremely perilous situation. Timely ultrasound-based diagnosis is crucial as it can enable conservative management with Methotrexate. Delayed diagnosis can lead to uterine rupture with consecutive surgery based on a transection of the pregnancy and cornual uterine resection.

Interstitial pregnancy (IP) is an infrequent variant of ectopic pregnancy (EP), with a documented occurrence rate of 2–3% [1]. It is characterized by the embedding of a blastocyst near the end of the fallopian tube, which penetrates the uterine myometrium [2]. The interstitial part of the fallopian tube, owing to its thickness and abundant vascular connections between the uterine and ovarian arteries, has a considerably higher potential for expansion. This can result in severe and life-threatening bleeding during rupture. The mortality rate associated with a ruptured interstitial ectopic pregnancy is approximately 2–5% [3]. The majority of ruptures in cases of interstitial ectopic pregnancies typically happen at 12 weeks of gestation. Therefore, early detection and intervention are of paramount importance for reducing substantial maternal health risks and mortality [4]. Nonetheless, diagnosing the clinical condition presents a distinct challenge as it is frequently mistaken for intrauterine pregnancy due to the significant expansion of the myometrium. This, in turn, leads to delayed treatment [3]. The prevalence of ectopic pregnancies is on the rise, and there is a noticeable simultaneous increase in the percentage of IPs. This increase in IP cases could be attributed to several factors, such as improved diagnostic methods, a higher occurrence of pelvic inflammatory disease, more frequent pelvic surgeries, and the use of assisted reproductive techniques [5]. An abnormal conceptus, impaired tubal passage, and altered tubal anatomy are factors in the etiology of EP [6]. Interstitial pregnancies account for approximately 2–6.8% of all ectopic pregnancies [5,7,8]. The key factors in the treatment of EP are the location and size of the pregnancy. Traditionally, EP was treated with hysterectomy or cornual wedge resection via laparotomy because of the high morbidity associated with myometrial surgery, risk of bleeding, and technically difficult surgical approach [9]. A conservative approach and minimally invasive surgery have been developed to avoid the negative outcomes of laparotomy [10]. However, the major disadvantages of conservative treatment are a need for prolonged follow-up and uncertainty regarding treatment success [11]. Conversely, hysteroscopic surgery and laparoscopy combined with Methotrexate have shown promising results [10,12]. Many reports have described a successful laparoscopic approach for IP in the first trimester [13,14,15,16]. However, in cases of hemoperitoneum and hemodynamic instability, many authors advocate a laparotomy approach [17].

Herein, we report a case of a successfully treated first-trimester IP associated with rupture and hemoperitoneum (Figure 1), with a diameter of 4 cm, via laparoscopic cornual wedge resection. A 36-year-old female with a history of ectopic pregnancy presented to our emergency department with lower abdominal pain. The patient denied vaginal bleeding. Medical history revealed an ectopic pregnancy that was operated on 10 years ago with a right salpingectomy. The quantitative beta-human chorionic gonadotropin (beta-hCG) level was >900 mIU/mL. Blood pressure was measured at 90/60 mmHg, and the heart rate was 110 beats per minute. A physical examination revealed moderate tenderness in the lower quadrants. A blood count showed anemia, 7.2 g/dL, and a hematocrit level of 33%. Ultrasonography disclosed a large quantity of free fluid in the abdomen and a mass on the right side of the uterus. Transfusion was performed, and the patient was referred to the OR.

We performed a laparoscopy and found approximately 3 L of blood and diagnosed a ruptured right interstitial ectopic pregnancy. Obstetrics were consulted, and a right cornual wedge resection via the laparoscopic approach was performed (Figure 2, Figure 3 and Figure 4). This technique is described as follows: informed consent was obtained from the patient for laparoscopy/laparotomy. As there was a high index of suspicion for ectopic pregnancy, we obtained consent for laparotomy and possible hysterectomy. In our case, there was no option for conservative management with Methotrexate and the watch-and-wait approach was considered not feasible. A Veress needle was used to create pneumoperitoneum. We used three trocars: 10 mm for the optic port placed umbilically and two working ports of 5 mm placed in the right and left flank, respectively. Operative findings revealed massive hemoperitoneum that was rapidly evacuated. Pelvic exploration revealed a normal left salpinx, ovary, and uterus, and an absent right salpinx and ovary. A 4/3 cm mass was noted in the right cornua with active bleeding.

The operative time was approximately 120 min. The specimen was analyzed, and all fetal parts were accounted for (Figure 5). A decrease in beta-hCG levels was noted postoperatively. The patient recovered uneventfully and was discharged on postoperative day four.

Cornual uterine tissue can stretch to allow pregnancy to progress to advanced gestation and hence remain undetected prior to rupture [4,7,11]. The rupture of anIP associated with proximity to intramyometrial arcuate vasculature leads to massive bleeding, which accounts for the increased mortality rate of IP compared to the median of all EPs (2–5% versus 0.2–0.5%) [5,11].

The classic triad in the diagnosis of EP, amenorrhea, abdominal pain, and bleeding, is found in less than 40% of IP cases [12]. To aid in the early diagnosis of IP, a high index of suspicion associated with raised serum beta-hCG and ultrasound or MRI arerequired [5]. Some authors advocate 3D ultrasonography, which allows for a more precise diagnosis [18]. In the present patient, there was intense abdominal pain and amenorrhea, but these were not associated with vaginal bleeding. Beta-hCG levels were high (>900 mIU/mL), and associated anemia and hypotension required urgent treatment. Ultrasound was an important tool in the diagnosis, in this particular case showing free fluid, hence raising the suspicion of hemoperitoneum. 

One particularity of IP is recurrence, especially on the same side. This risk is increased for all Ips, even if good conservative surgery is performed [19]. This is consistent with our case, which was a recurrent ectopic pregnancy. Assisted conception and tubal pathology may contribute to this risk [20]. 

The conservative approach is based on systemic Methotrexate in either a single- or multiple-dose regimen, and many studies have reported the successful treatment of early first-trimester IP [9,12]. In addition, local treatment with Methotrexate or potassium chloride has been described in conjunction with the surgical removal of the EP [21].

Medical treatment indications include multiple previous surgeries, extensive pelvic adhesions, contraindications for general anesthesia, cornual pregnancy, and the failure of conservative laparoscopic treatment [22]. Our case had a clinical picture and investigations of an emergency scenario; hence, conservative treatment with Methotrexate was not applied.

Indications for surgical treatment include a ruptured pregnancy, anemia, low blood pressure, a gestational sac diameter >4 cm on ultrasound, or lower abdominal/pelvic pain that persists for more than 24 h [23]. Moreover, unstable vital signs with free fluid on abdominal ultrasonography are an indicator for resuscitation and emergency operative management. In our case, the patient had clear indications for immediate surgery.

Surgical management of IP can be open or laparoscopic and includes cornual wedge resection, cornuostomy with removal of the gestational sac and pregnancy contents, or hysterectomy [11]. The SOGC Guideline recommends either laparoscopic cornuostomy or cornual wedge resection; both procedures have comparable results, but this definition has a low quality of evidence [24].

In a series of 11 consecutive cornual EPs, MacRae et al. reported cornual resection in seven (70%) [25]. Selma Ng et al. reported a series of 52 cases managed laparoscopically, of which 33 patients (62%) underwent wedge resection, 13 patients underwent cornuostomy, and 7 patients underwent salpingectomy [26].

In addition, the hysteroscopic removal of an EP using a laparoscope-assisted approach has been described [13]. We opted for conservative surgery based on wedge resection with satisfactory results in our patient.

Uterine rupture is a major complication of an IP. Many authors prefer laparotomy in emergency scenarios such as a ruptured IP, especially after the first trimester [14].

Hysterectomy, a radical surgery, is indicated in case of hemorrhage that cannot be controlled/is profuse [15,27].

Laparoscopic management has multiple advantages: less postoperative pain, less postoperative analgesia, shorter hospitalization, and a faster return to work. Reduced blood loss and wound complications are also observed [6,28].

De Cherney et al. reported the first large American series of 79 cases of ectopic pregnancy being successfully treated laparoscopically [16]. Since then, the use of laparoscopy for the treatment of EP has increased, replacing laparotomy in many centers [28,29].

One problem related to the laparoscopic approach is the length of surgery. Cornual wedge resection involves tying multiple knots, which is time-consuming [30]. Due to our experience in laparoscopic surgery, we managed to finalize the intervention in approximately 2 h. 

Another aspect can be related to the size of the EP, which ranges from <1 to 6 cm, corresponding to the age of the EP. The EP in our case had a diameter of 4 cm, which is close to the mean reported in the literature of approximately 3 cm [26].

Technical difficulties in the laparoscopic approach can lead to conversion. In a series of twentypatients primarily treated by laparoscopy, six (30%) underwent conversion to laparotomy: two cases were related to hemoperitoneum, and four weredue to technical difficulty [31].

A systematic review suggested that laparoscopy should be the first-choice method when a surgical approach is required in patients diagnosed with an IP [32]. Our experience is similar to the above review, and we extend this indication to a ruptured EP, particularly an IP. One technical difficulty with aruptured IP is managing profuse bleeding of the myometrium, but using energy devices such as Ligasure can prove useful, and absorbable hemostat aids in hemostasis. Postoperative beta-hCG level follow-up is recommended after surgery for an IP as this is a marker of non-persistent IP tissue. There is a recommendation for Methotrexate injection after surgery if persistently high serum beta-hCG levels are present [33]. There was a significant decrease in beta-hCG levels (fourfold less than the initial values); therefore, we did not recommend Methotrexate for the patient presented herein.

A ruptured ectopic pregnancycan affect the immediate health of the woman, but there are multiple studies showing that ectopic pregnancy does not significantly decrease the intrauterine pregnancy rate nor increase thefuture ectopic pregnancy rate during the 12-month follow-up period [34,35,36,37].

Early diagnosis and prompt treatment are paramount in the management of interstitial pregnancies. The history of the patient and elevated serum beta-hCG levels associated with ultrasound can aid in diagnosis. In cases of rupture and intraperitoneal bleeding, immediate surgical intervention is mandatory. The peculiarity of the case presented here offers an example of a laparoscopic approach in an emergency scenario in a patient with hemoperitoneum and a ruptured interstitial pregnancy. Therefore, laparoscopy can be regarded as a safe and effective treatment for ruptured interstitial pregnancy.

## Figures and Tables

**Figure 1 diagnostics-14-00506-f001:**
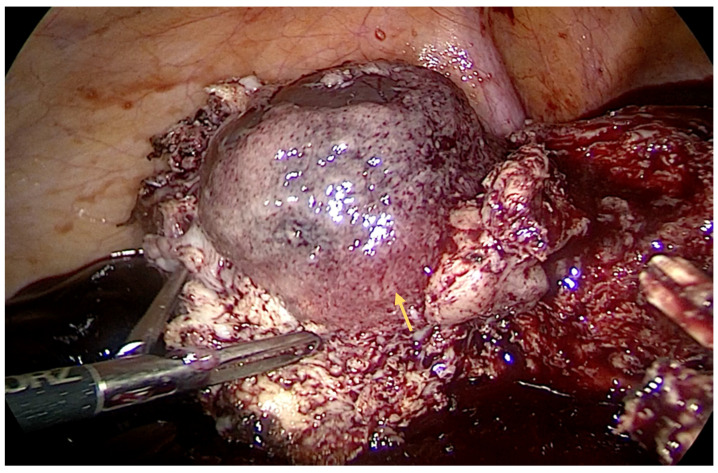
Intraoperative aspect—interstitial pregnancy: enlarged mass in right cornua (indicated by yellow arrow).

**Figure 2 diagnostics-14-00506-f002:**
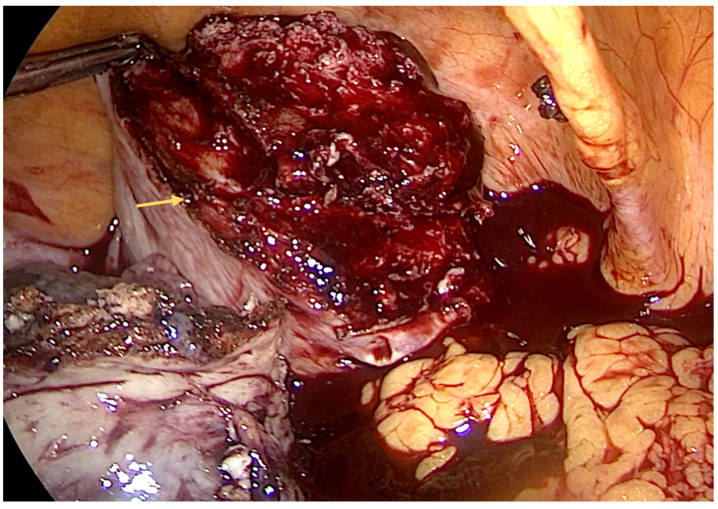
Intraoperative aspect—the interstitial pregnancy transected off the uterus (indicated by a yellow arrow). The serosa underlying the mass was hyperemic. Using hook electrocautery, we performed a dissection of the mass (interstitial pregnancy), and the transection was finalized using Ligasure^®^.

**Figure 3 diagnostics-14-00506-f003:**
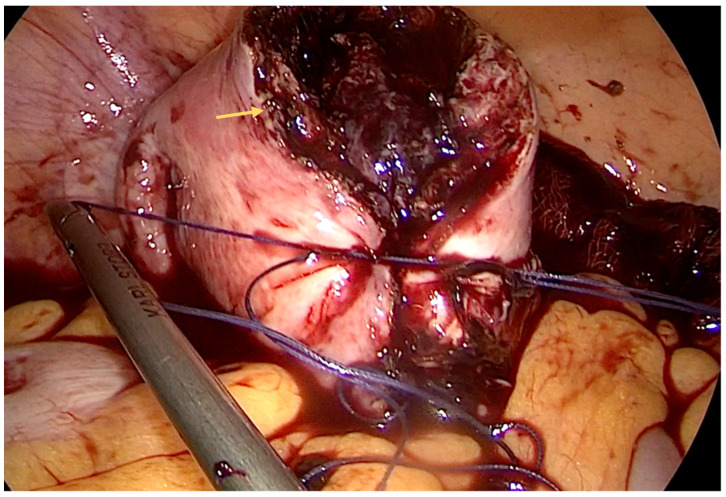
Intraoperative aspect—several resorbable sutures were used to reapproximate the uterine myometrium (indicated by a yellow arrow). Hemostasis was achieved with electrocautery and using a Pahacel^®^ absorbable hemostat (oxidized regenerated cellulose). The myometrial defect was reapproximated with a pair of 0 absorbable (Vicryl) sutures using the laparoscopic standard suture technique.

**Figure 4 diagnostics-14-00506-f004:**
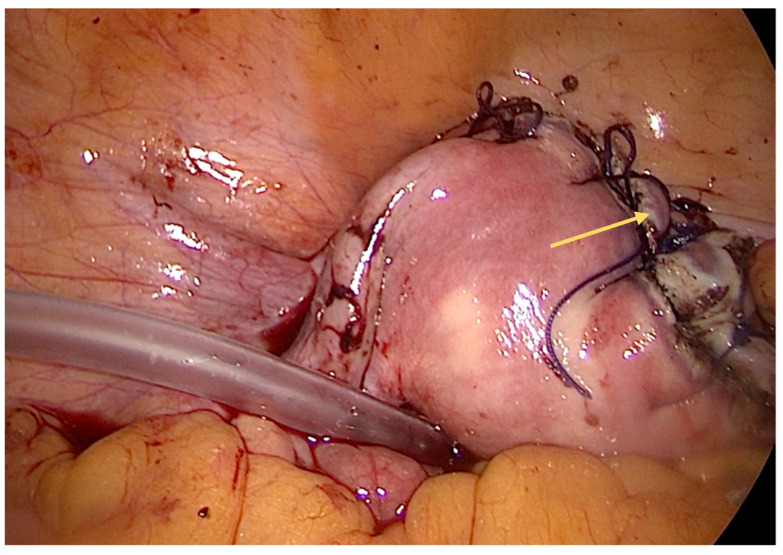
Intraoperative aspect—cornual resection performed and laparoscopic closure of the uterus (indicated by yellow arrow), with a retro-uterine drain placed. The abdomen was washed with a saline solution, and a drain was placed in the retro-uterine space.

**Figure 5 diagnostics-14-00506-f005:**
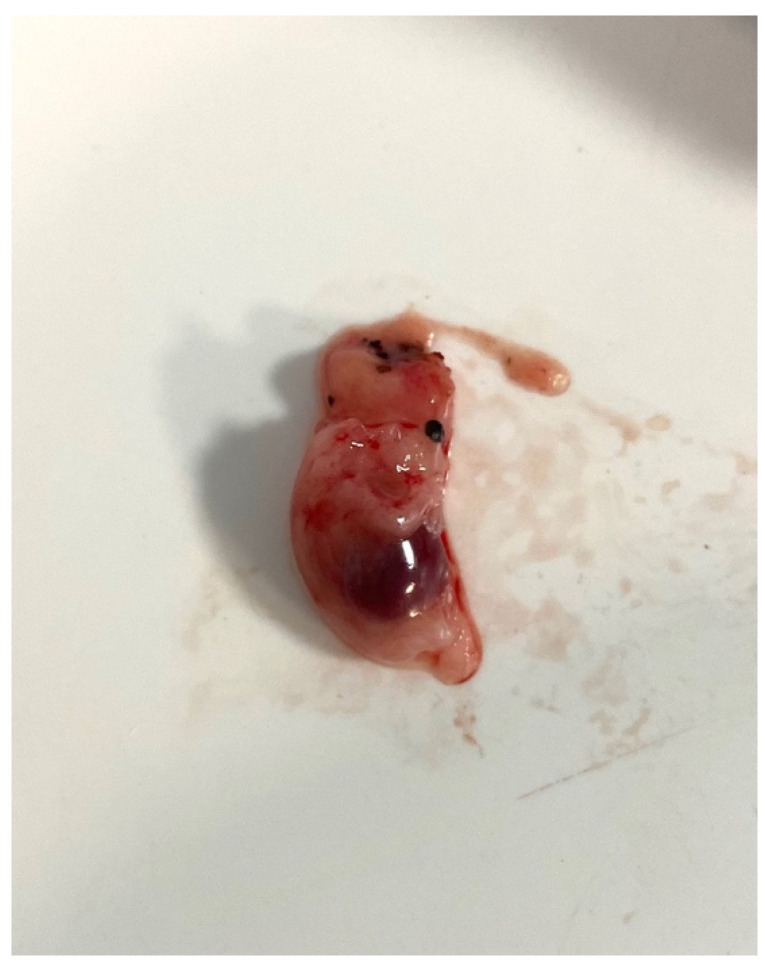
The specimen—all fetal parts wereaccounted for—was removed from theabdomen using an Endobag through theumbilical incision.

## Data Availability

All data generated or analyzed during the present study are included in this published article.

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
