# Peer review of "Ruptured Recurrent Interstitial Ectopic Pregnancy Successfully Managed by Laparoscopy"

_diagnostics, 2024, doi:10.3390/diagnostics14050506_

Round 1

Reviewer 1 Report

Comments and Suggestions for Authors

The case presentation is interesting as interstitial pregnancy is a rare occurrence, and the previous history of homolateral ectopic pregnancy is also an interesting coincidence.

The authors describe the case, its management and particularities in a clear manner. The intraoperative images are also very nice.

I would recommend that the authors insert some discussions also on this particular succession of pathologies, details on the diagnostic protocol and future reproductive prognosis.

Also, an issue that needs to be addressed is the lack of the ethical authority.

Author Response

Dear Reviewer 1,

Thank you for your kind words. We have presented our ethical consent (Approval code: 9089, Date of approval: 12.04.2023). Written informed consent was obtained from the patient.

We have also introduced more data regarding future reproductive prognosis in the revised manuscript.

We look forward to further suggestions if needed

Warm Regards

Reviewer 2 Report

Comments and Suggestions for Authors

The case report submitted for review concerns an important aspect of ectopic pregnancy located in the intramural part of the fallopian tube. However, the manuscript provides only basic and obvious knowledge on this topic. Not suitable for publication in Diagnostics.

Comments on the Quality of English Language

 Minor editing of English language required

Author Response

Dear Reviewer 2,

We respect your opinion. However, the case report presented shows an important aspect of the treatment for ruptured ectopic pregnancies: the laparoscopic approach. The literature is inconclusive on this aspect, and many authors recommend an open approach. We believe that our case report encourages a minimally invasive technique. Although our presentation may seem obvious and depict some basic knowledge, I believe that many clinicians and surgeons are interested in the diagnosis and treatment of this important and life-threatening pathology in women. 

Reviewer 3 Report

Comments and Suggestions for Authors

Comments on the Quality of English Language

Author Response

Thank you for your comments and suggestions. In the revised form of the manuscript all the corrections have been made accordingly to your pertinent observations.

Reviewer 4 Report

Comments and Suggestions for Authors

Dear colleagues!

It is not obvious to me that the clinical case is placed in the Interesting images section.

I recommend that you read the procedure for registering a clinical case. The quality of the images is not high, so it is advisable to mark the areas you are describing with markers.

As you know, the laparoscopic surgical technique for ectopic pregnancy is quite successful and you need to point out the advantages of the method you have chosen.

Author Response

Dear Reviewer,

Thank you for your comment. We found that our images present an important aspect of the laparoscopic approach for ruptured ectopic pregnancy, and hence, our choice for this section. In the revised version of the manuscript we have made changes to the images accordingly to your suggestions.

Despite the well known advantages of the minimally invasive approach, many authors still recommend an open surgery in this cases in favor of laparoscopy. The literature is inconclusive, and RCT-s are needed to state a definitive approach.

Warm regards

Round 2

Reviewer 2 Report

Comments and Suggestions for Authors

I'm offering my opinion again. I have no objections to the form and content of the manuscript. In this respect, it can be published. My reservations concern only the rather obvious aspects described in this case report. For me personally, it is not worth reading, but maybe some readers will be happy to read it. I leave the final decision to the publisher.